# The Pharmaceutical Industry in 2018. An Analysis of FDA Drug Approvals from the Perspective of Molecules

**DOI:** 10.3390/molecules24040809

**Published:** 2019-02-23

**Authors:** Beatriz G. de la Torre, Fernando Albericio

**Affiliations:** 1KRISP, School of Laboratory of Medicine and Medical Science, College of Health Sciences, University of KwaZulu-Natal, Durban 4001, South Africa; 2School of Chemistry and Physics, University of KwaZulu-Natal, Durban 4001, South Africa; 3CIBER-BBN, Networking Centre on Bioengineering, Biomaterials and Nanomedicine, Department of Organic Chemistry, University of Barcelona, 08028 Barcelona, Spain

**Keywords:** antibodies, API, biologics, chemical entities, drug discovery, fluorine based drugs, natural products, oligonucleotides, peptide, TIDES, small molecules

## Abstract

The Food and Drug Administration approved 59 new drugs (42 New Chemical Entities and 17 Biologics) during 2018. This number breaks the previous record of 53 approved by the same organization in 1996. The 17 new biologics approved in 2018 also represent an important milestone for this kind of drug and they clearly exceed the 12 approved in 2015 and 2017. Herein, the 59 new drugs of the class of 2018 are analyzed from a strictly chemical perspective. The classification has been carried out on the basis of the chemical structure and includes the following: Biologics (antibodies and enzymes); TIDES (peptides and oligonucleotides) and natural products; drug combinations; and small molecules.

## 1. Analysis

2018 stands out as a year in which the number of new drugs approved by the Food and Drug Administration (FDA) broke a record. In this regard, 59 new drugs (42 New Chemical Entities (NCE) and 17 Biologics) were approved, exceeding the 53 authorized in 1996 (47 NCEs and 6 Biologics) [1]. The numbers in 2018 are a continuation of the previous year, which witnessed the approval of 46 new entities (34 + 12) [1,2,3] (Figure 1). Will this increasing trend of both kinds of drug continue in coming years? Analysts are cautious in responding to this question since the approval of a new drug by the corresponding agencies involves many variables that are difficult to predict [4,5]. Even for this year (2019), the current shutdown of the United States government could have a negative impact on the final number of approved drugs if it lasts for an extended period of time.

Taking biologics into account, the 17 approved in 2018 also represent a record, clearly surpassing the 12 approved in 2015 and 2017. These numbers are a confirmation of the increasing importance of these kinds of pharmaceutical drug, which in the last five years (2014–2018) account for more than 25% (59 of 213) of all drugs approved, and therefore the consolidation of these molecules. In contrast to the excellent performance of biologics, fewer new products were approved by the Center for Biologics Evaluation and Research (CBER) in 2018 than in 2017 (3 vs. 6) [4,6]. However, in this regard, it is important to highlight the approval of the first hexavalent vaccine.

## 2. Discussion

Table 1 shows the 17 biologics approved in 2018, of which 12 are monoclonal antibodies (mAb), three pegylated enzymes, one protein, and one fusion protein (Table 1).

Once again, mAb are the most important class of biologics and even of drugs. In 2018, 11 mAb were approved, which account for almost 20% of all drugs authorized by the FDA in 2018. This number exceeds those registered in 2016 and 2017 (7 and 9, respectively), thus leading to the approval of 27 antibody-based drugs out of a total of 127 new drugs over these three years. Interestingly, three drugs Erenumab, Fremanezumab, and Galcanezumab are indicated for the migraine prevention.

It is important to draw attention to the approval of three pegylated enzymes in 2018. After the approval of the highly pegylated peptide peginesatide (Omontys^TM^) by the FDA in 2012 and its later withdrawal from the market a few months later, it appeared that the pharmaceutical sector was somewhat reluctant to introduce polyethylenglycol (PEG) moieties into their drug discovery programs. The acceptance of three pegylated drugs in the same year is expected to once again strengthen the development of PEG-containing drugs.

After the approval of trastuzumab emtansine (Kadcyla^TM^) in 2013 and inotuzumab ozogamicin (Besponsa^TM^) in 2017—both antibody drug conjugates (ADCs), 2018 witnessed the approval of two drugs based on the same idea but with a different chemical construction to that of the ADCs. Thus, moxetumomab pasudotox (Lumoxiti^TM^) is a recombinant immunotoxin formed by an antibody covalently bound to a fragment of Pseudomonas exotoxin-A. On the other hand, tagraxofusp-erzs (Elzonris^TM^) is an interleukin 3-based fusion protein containing the diphtheria toxin. Analysts forecast that more ADCs or fusion proteins containing toxins will be approved by the FDA in the coming years, as reflected by the large clinical pipeline of this kind of compounds [7].

TIDES (oligonucleo- and pep-TIDES), which are prepared chemically, stand among biologics and the so-called small molecules, which are characterized by their tendency to follow the Lipinski rules. In this regard, 2018 showed a similar trend to 2016, in which three oligonucleotides and one peptide were approved [8,9]. The approval of patisiran (Onpattro^TM^) for the treatment of hereditary transthyretin-mediated amyloidosis in adults could probably be considered the most impressive breakthrough of recent years in the drug discovery arena. Patisiran, which is a double-stranded short interfering RNA (siRNA) encapsulated in a lipid nanoparticle, has met expectations after the discovery of the RNAi pathway in 1998 (Figure 2). On this occasion, a reasonable period of 20 years has elapsed between the publication of the first seminal paper and the first siRNA drug to reach the market. Other RNAi therapeutics are in clinical phases and will presumably be approved in the coming years, thus extending the medical applications of this technology. For the treatment of the same disorder, inotersen (Tegsedi^TM^) (Figure 2), which is a single-stranded phosphorothioate oligonucleotide that acts through an antisense mechanism, was also approved in 2018. The five oligonucleotide-based drugs approved in 2016 and 2018 are starting to give returns on the large investments made by the pharmaceutical industry in this field since the earlier 1990s and consolidate these chemical species as an important class of drugs.

With regard to peptides, 2018 has witnessed the approval of Lutetium Lu 177 DOTA-TATE (Lutathera^TM^), which is used for PET imaging and peptide receptor radionuclide therapy. Lu 177 DOTA-TATE is composed by the cyclic octapeptide (Tyr^3^-octeotride) terminated at the *N*-terminal with a DOTA chelator, to which Lu-177 is bound (Figure 3). Lu 177 DOTA-TATE is used for the diagnosis and treatment of neuroendocrine tumors, which overexpress somatostatin receptors and therefore show a high affinity for Tyr^3^-octeotride. Similar theranostics are expected to be accepted by the agencies in the coming years.

In the section of drugs inspired in natural products, 2018 has been a superb year, with 10 drugs approved, thus revalorizing once again the important role of natural products in drug discovery.

First of all, it is important to highlight three tetracycline antibiotics for several infectious disorders (Figure 4). In this regard, eravacycline (Xerava^TM^), which is indicated for severe intra-abdominal infections, contains one atom of F (see above for comments about the large number of drugs approved containing F; F is indicated in green in all structures). Omadacycline (Nuzyra^TM^) was approved for the treatment of community-acquired bacterial pneumonia (CABP) and acute bacterial skin and skin structure infections (ABSSSI). Finally, sarecycline (Seysara^TM^), which is the simplest of the three tetracyclines, is recommended for the treatment of moderate-severe acne vulgaris.

Two carbohydrate-inspired drugs were also approved—migalastat (Galafold^TM^) and plazomicin (Zemdri^TM^) (Figure 5). The former is an iminosugar with a stereochemistry similar to the D-galactose derived from the natural product njirimycin, while the latter is an aminoglycoside derived from sisomicin, which is turn is derived from gentamicin. Migalstat is used to treat Fabry disease and plazomicin to treat infections of the urinary tract.

Two macrocycles, moxidectin (Moxidectin^TM^) and rifamycin SV (Aemcolo^TM^) were approved in 2018 (Figure 6). The former is derived from nemadectin and was first used as an antiparasitic in veterinary medicine. It was approved by the FDA for the treatment of onchocerciasis (river blindness). Rifamycin SV belongs to a group of antibiotics that are effective against mycobacteria, but that has been approved for the treatment of travelers’ diarrhea.

With regard to the steroid family, this year has seen the approval of Annovera^TM^ (Figure 7), which is a combination of two steroids, namely the estrogen ethinylestradiol and the progestin segesterone acetate. This drug is indicated as a contraceptive vaginal ring.

2018 also witnessed the approval of the first marijuana-derived drug by the FDA (Figure 8). Cannabidiol (Epidiolex^TM^) is indicated for the treatment of the syndromes of Dravet and Lennox–Gastaut, which are two severe forms of epilepsy in infancy.

Finally, Omegaven^TM^, which is a fatty acid emulsion derived from fish oil and very rich in omega-3, was approved for parenteral nutrition associated with cholestasis, which describes a condition in which bile cannot flow from the liver to the duodenum.

In addition to Annovera^TM^, three other drug combinations were approved in 2018, thereby confirming the tendency in recent years for these drugs to contain more than one API.

Biktarvy^TM^ is a combination of bictegravir, emtricitabinem and tenofovir alafenamide in a single tablet for the treatment of HIV-1 infection (Figure 9).

Akynzeo^TM^ is a fixed dose combination of palonosetron and fosnetupitant for the treatment of acute nausea and vomiting produced by chemotherapy. Again, it is interesting to highlight that fosnetupitant has two trifluoromethyl groups in its structure (Figure 10).

Symdeko^TM^, which has been approved for the treatment of cystic fibrosis, is a combination of tezacaftor and ivacaftor (Figure 11). The structure of the former also contains several F.

In addition to eravacycline, bictegravir and emtricitabine –part of Biktarvy^TM^–, fosnetupitant –part of Akynzeo^TM^–, and tezacaftor–part of Symdeko^TM^–, another 14 drugs contain F. This implies that almost one third (18 over 59) of all drugs approved by the FDA during 2018 contain F. Taking into account the NCE alone, this proportion increases to slightly more than 40% (18 vs. 42).

In addition to fosnetupitant, five more drugs with trifluoromethyl groups were approved in 2018. Apalutamide (Erleada^TM^), which contains a trifluoromethyl pyridine and an additional fluorophenyl moiety, is indicated for the treatment of non-metastatic castration-resistant prostate cancer. Apalutamide could be considered the “n” generation of flutamide (Eulexin^TM^), which also contains the trifluoromethyl group and is indicated for the treatment of prostate cancer (Figure 12).

Elagolix sodium (Orilissa^TM^), which contains one trifluoromethyl group and two additional fluoroaryl moieties, was approved for the treatment of pain associated with endometriosis (Figure 13).

Finally, doravirine (Pifeltro^TM^), tafenoquine (Krintafel^TM^), and tecovirimat (TPOXX^TM^) contain just a single trifluoromethyl group and are indicated for the treatment of HIV/AIDS, Plasmodium vivax malaria, and smallpox, respectively (Figure 14). Tafenoquine is a mixture of two enantiomers.

Nine more drugs approved in 2018 contain fluoroaryl moieties and in all cases nitrogen-based heterocycles. The presence of these drugs is also a highlight of the year. This is probably due to the large number of so-called small molecules that gained approval.

In spite of having an unrelated chemical structure, binimetinib (Mektovi^TM^) and encorafenib (Braftovi^TM^) were approved for the treatment of BRAF-mutated melanoma (Figure 15).

Two drugs for non-small-cell lung carcinoma (NSCLC) were approved, namely dacomitinib (Vizimpro^TM^) for EGRF-mutated NSCLC and lorlatinib (Lorbrena^TM^) for ALK-positive metastatic NSCLC (Figure 16). The latter is 12-member medium macrocycle with three aryl moieties.

The rest of molecules approved in 2018 having F were baloxavir marboxil (Xofluza^TM^) for the treatment of influenza A and B; fostamatinib (Tavalisse^TM^), which contains a phosphate, for chronic immune thrombocytopenia (ITP); ivosidenib (Tibsovo^TM^), which also contains a difluorociclobutyl, for acute myeloid leukemia and cholangiocarcinoma; larotrectinib (Vitrakvi^TM^) for NTRK gene fusion–positive solid tumors; and talazoparib (Talzenna^TM^) for BRCA-mutated HER2-negative breast cancer (Figure 17).

Two thiazol-2-amide containing molecules, namely avatrombopag (Doptelet^TM^) and lusutrombopag (Mulpleta^TM^) (Figure 18), were approved for the treatment of thrombocytopenia, which is characterized by abnormally low platelet counts and associated with chronic liver disease. Both show related chemical structures.

Two drugs with unrelated chemical structures, glasdegib (Daurismo^TM^) and gilteritinib (Xospata^TM^) (Figure 19), received approval for the treatment also of acute myeloid leukemia. Including ivosidenib (see above), three drugs were approved for this indication in 2018.

The following drugs with nitrogen-based aromatic heterocycles were also approved last year: baricitinib (Olumiant^TM^) for rheumatoid arthritis; duvelisib (Copiktra^TM^) for various hematologic malignancies such as chronic lymphocytic, small lymphocytic, and follicular lymphomas; and amifampridine (Firdapse^TM^), which just is 3,4-diaminopyridine, for Lambert–Eaton myasthenic syndrome, which is a muscle disease (Figure 20). In 2010, the FDA also approved 4-diaminopyridine Ampyra^TM^ to improve walking in adults with multiple sclerosis.

Four more small-molecule drugs were approved in 2018, namely stiripentol (Diacomit^TM^) for Dravet syndrome, which is an intractable infancy epilepsy, lofexidine (Lucemyra^TM^) for the management of the physical symptoms of opioid withdrawal, prucalopride (Motegrity^TM^) for chronic idiopathic constipation, and revefenacin (Yupelri^TM^) for chronic obstructive pulmonary disease (COPD). Lofexidine is a racemic mixture (Figure 21).

Finally, the FDA approved an inorganic compound, sodium zirconium cyclosilicate (Lokelma^TM^) [(2Na·H_2_O·3H_4_SiO_4_·H_4_ZrO_6_)_n_], an oral sorbent for potassium ions throughout the gastrointestinal tract used for the treatment of hyperkalemia, which is characterized by elevated levels of serum potassium.

## 3. Conclusions

Figure 22 shows the drugs approved by the FDA in 2018 and classified on the basis of their chemical structure.

Although it is widely recognized that trends and tendencies in “Drugs to the Market” are difficult to analyze and even more difficult to forecast for the coming years, it is clear that 2018, together with previous years, indicate the dominance of antibodies as the “Drugs of the Future”. In this regard, 11 mAb and an additional one covalently bound to a fragment of an exotoxin were approved in 2018. Furthermore, and although no ADC has been approved to date, two macromolecules bound to toxin fragments were approved, thus reaffirming the concept of ADCs [8]. Another highlight regarding biologics has been the reinstatement of pegylation as a tool in drug discovery.

From a strictly chemical point of view, the “Molecule of the Year” was probably patisiran (Onpattro^TM^). This drug marks the approval of the first double-stranded siRNA and therefore opens the door for the authorization of others. A second oligonucleotide also entered the market in this reporting year, which may indicate the consolidation of these kinds of molecules as drugs.

From a quantitative perspective, it is important to highlight the 10 drugs inspired in natural products. Although such compounds always have critics, their relevance for the discovery of new drugs is unquestionable. Cannabidiol (Epidiolex^TM^) has been the first marijuana-derived drug approved by the FDA.

Fluorine deserves nomination as the “Atom of the Year”, not only because three out of ten drugs of the 2018 class contain this atom, but because a total of 49 F are present in the 18 fluorine-containing drugs.

Most small molecule-based drugs contain nitrogen-based aromatic heterocycles such as pyrazoles, imidazoles, benzoimidazoles, triazoles, and 2-amine-pyrimidine moieties.

As in 2017 [9,10], oncology drugs received the most approvals in 2018, but were followed very closely by drugs indicated for the treatment of infectious diseases. The efforts of the pharmaceutical industry to tackle these diseases are paying off, as reflected by the large number of drugs (14) accepted this year.

Although 2018 was overall a terrific year in terms of the number of drugs accepted by the FDA, it is important to highlight the increase in the cost of treatments. For many of these new drugs, the yearly cost is estimated to reach a six-digit figure. Consequently, these drugs will be unaffordable for most of the population, even in developed countries with excellent medical care programs. This is a message to take home.

## Figures and Tables

**Figure 1 molecules-24-00809-f001:**
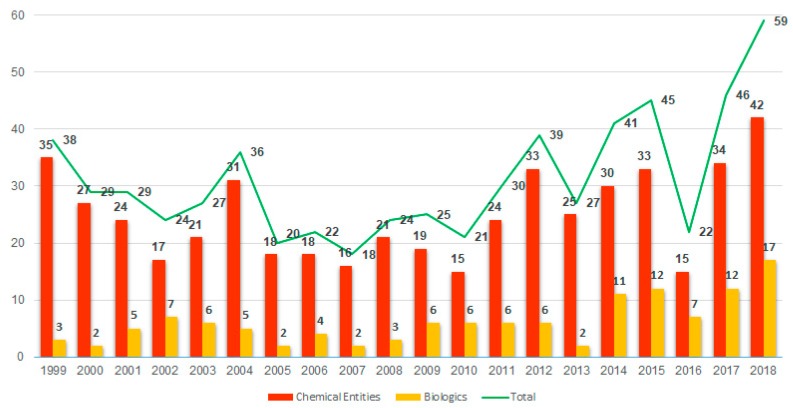
New chemical entities and biologics approved by the FDA in the last two decades [1,4,5].

**Figure 2 molecules-24-00809-f002:**
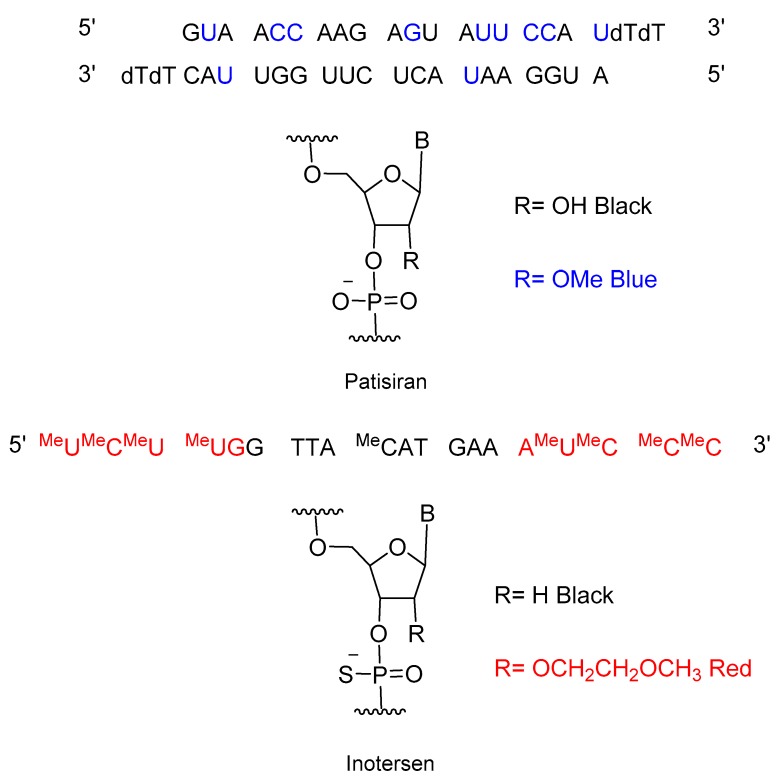
Structure of oligonucleotide based drugs, patisiran and inotersen.

**Figure 3 molecules-24-00809-f003:**
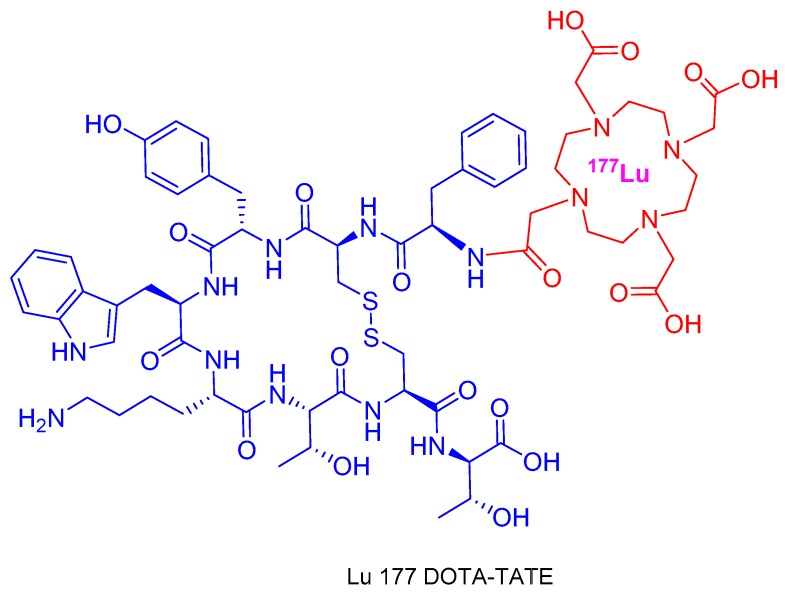
Structure of Lu 177 DOTA-TATE.

**Figure 4 molecules-24-00809-f004:**
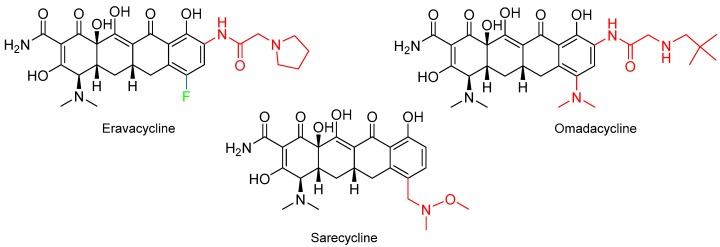
Structures of tetracycline drugs. In red the variable part.

**Figure 5 molecules-24-00809-f005:**
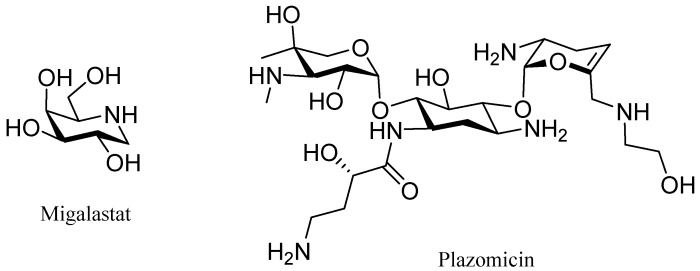
Structure of carbohydrate derived drugs.

**Figure 6 molecules-24-00809-f006:**
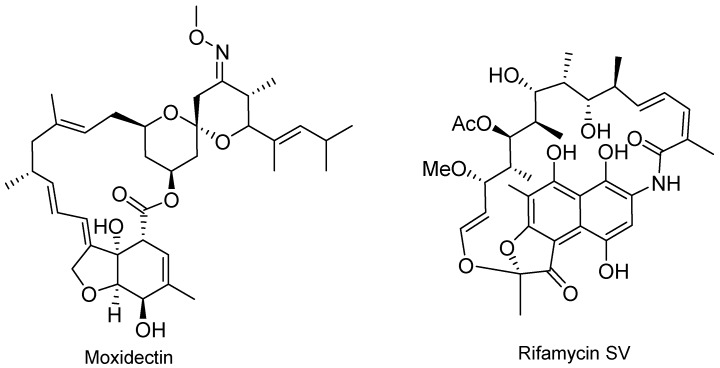
Structure of drugs containing a macrocycle.

**Figure 7 molecules-24-00809-f007:**
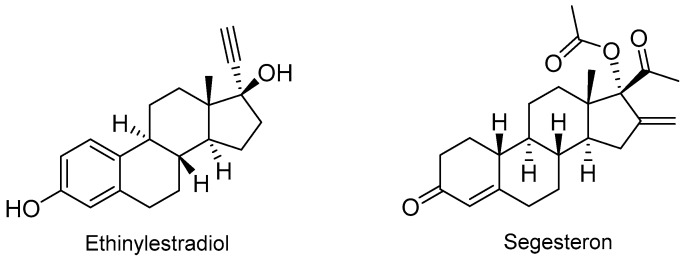
Structure of Annovera^TM^, a combined steroid derived drug.

**Figure 8 molecules-24-00809-f008:**
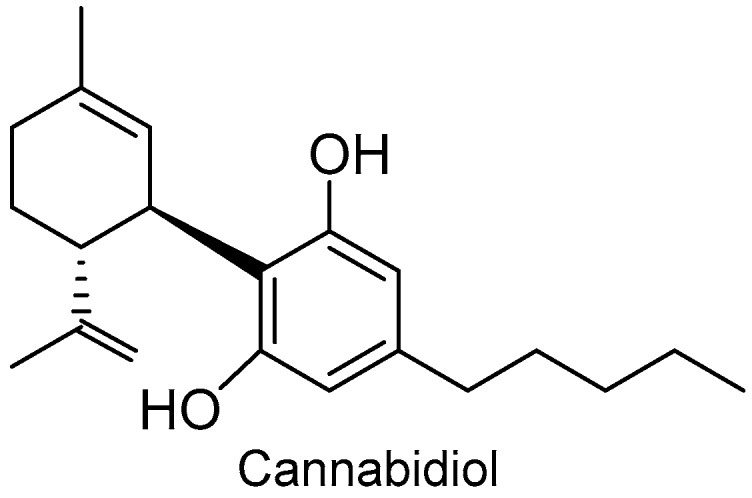
Structure of cannabidiol.

**Figure 9 molecules-24-00809-f009:**
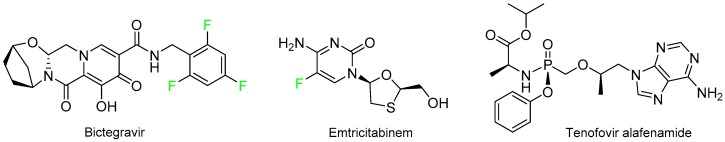
Structure of Biktarvy^TM^, a drug combination.

**Figure 10 molecules-24-00809-f010:**
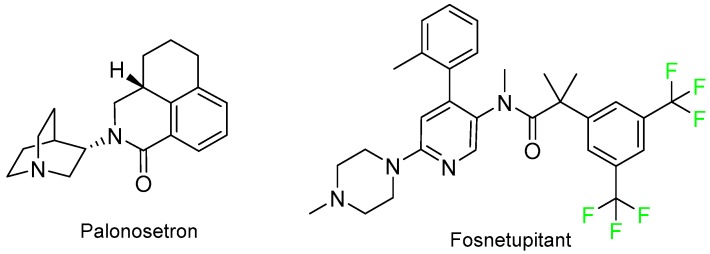
Structure of Akynzeo^TM^, a drug combination.

**Figure 11 molecules-24-00809-f011:**
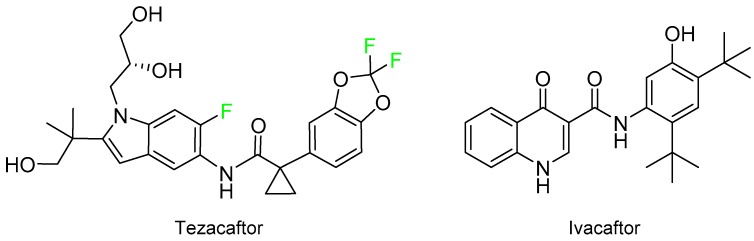
Structure of Symdeko^TM^, a drug combination.

**Figure 12 molecules-24-00809-f012:**
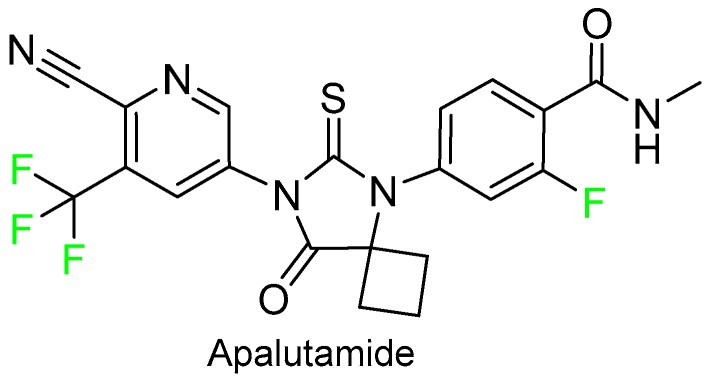
Structure of apalutamide.

**Figure 13 molecules-24-00809-f013:**
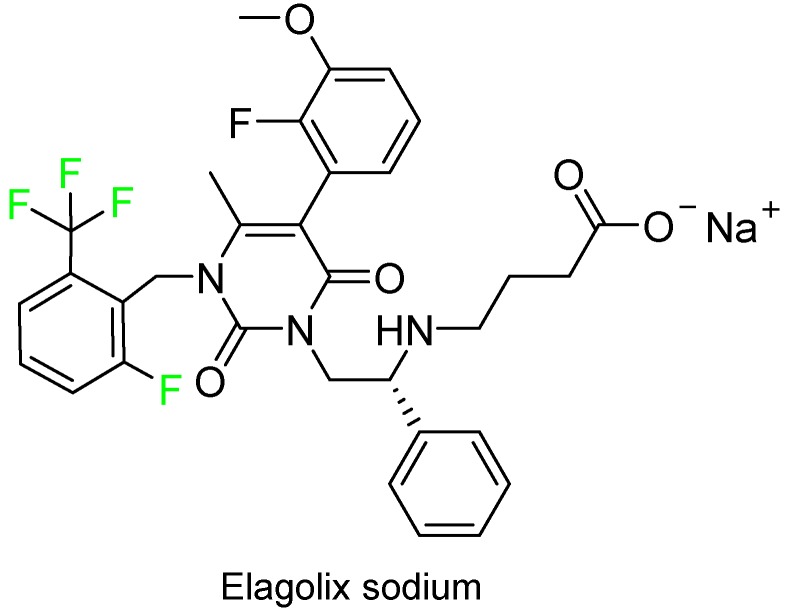
Structure of elagolix sodium.

**Figure 14 molecules-24-00809-f014:**
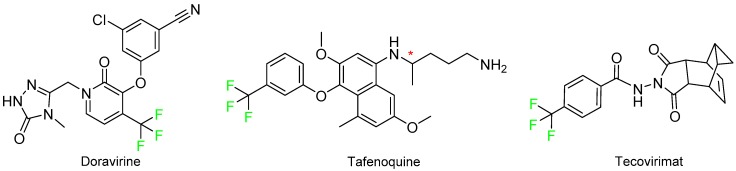
Structure of doravirine, tafenoquine, and tecovirimat (* denotes a chiral center).

**Figure 15 molecules-24-00809-f015:**
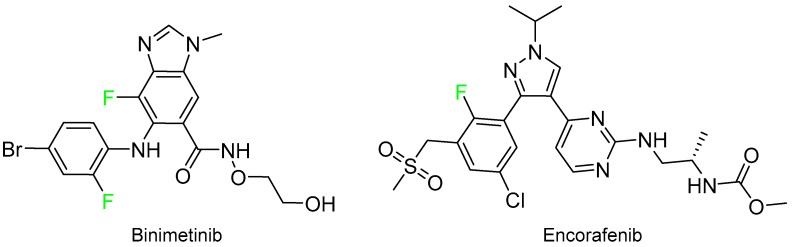
Structure of drugs for BRAF-mutated melanoma.

**Figure 16 molecules-24-00809-f016:**
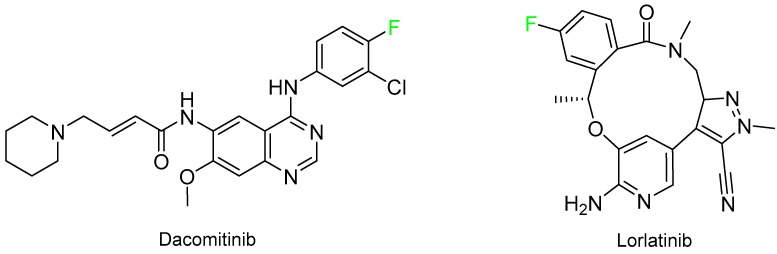
Structure of drugs for NSCLC.

**Figure 17 molecules-24-00809-f017:**
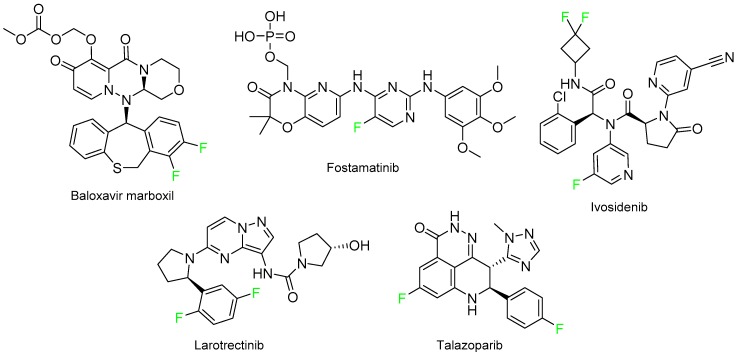
Structure of drugs containing fluoroaryl moieties.

**Figure 18 molecules-24-00809-f018:**
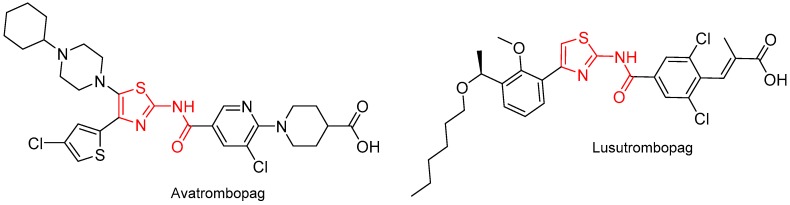
Structure of thiazol-2-amide (in red) containing molecules.

**Figure 19 molecules-24-00809-f019:**
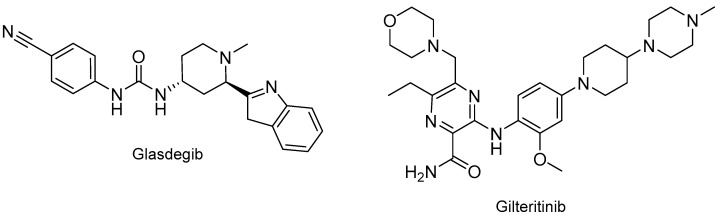
Structure of drugs for the treatment of acute myeloid leukemia.

**Figure 20 molecules-24-00809-f020:**
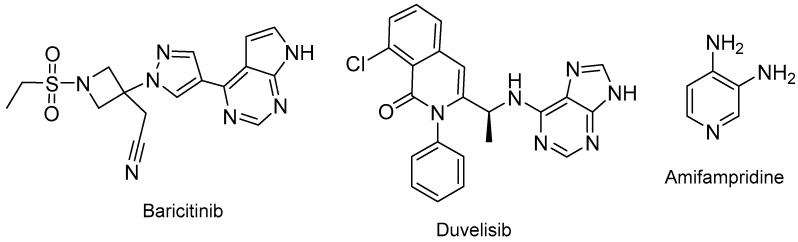
Structure of baricitinib, duvelisib, and amifampridine.

**Figure 21 molecules-24-00809-f021:**
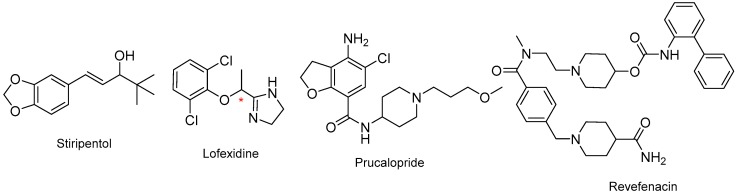
Structure of stiripentol, lofexidine, prucalopride, and revefenacin. (* denotes a chiral center).

**Figure 22 molecules-24-00809-f022:**
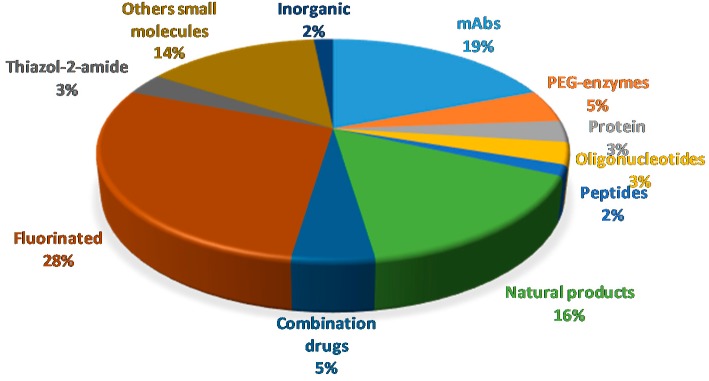
Drugs approved by the FDA in 2018 and classified on the basis of their chemical structure.

**Table 1 molecules-24-00809-t001:** Biologics approved by the FDA in 2018 [1].

Active Ingredient ^a^	Trade Name ^b^	Class	Disease
Burosumab	Crysvita^TM^	monolclonal antibody	X-linked dominant hypophosphatemic rickets
Calaspargase pegol	Asparlas^TM^	pegylated enzyme	Acute lymphoblastic leukemia
Cemiplimab	Libtayo^TM^	monolclonal antibody	Cutaneous squamous cell carcinoma
Cenegermin	Oxervate^TM^	protein	Neurotrophic keratitis
Elapegademase	Revcovi^TM^	pegylated enzyme	Adenosine deaminase severe combined immunodeficiency
Emapalumab	Gamifant^TM^	monolclonal antibody	Hemophagocytic lymphohistiocytosis
Erenumab	Aimovig^TM^	monolclonal antibody	Migraine prevention
Fremanezumab	Ajovy^TM^	monolclonal antibody	Migraine prevention
Galcanezumab	Emgality^TM^	monolclonal antibody	Migraine prevention
Ibalizumab	Trogarzo^TM^	monolclonal antibody	Multidrug-resistant HIV-1
Lanadelumab	Takhzyro^TM^	monolclonal antibody	Hereditary angioedema attacks
Mogamulizumab	Poteligeo^TM^	monolclonal antibody	Relapsed or refractory mycosis fungoides and Sézary disease
Moxetumomab pasudotox	Lumoxiti^TM^	monolclonal antibody	Relapsed or refractory hairy cell leukemia
Pegvaliase	Palynziq^TM^	pegylated enzyme	Phenylketonuria
Ravulizumab	Ultomiris^TM^	monolclonal antibody	Paroxysmal nocturnal hemoglobinuria and atypical hemolytic uremic syndrome
Tagraxofusp-erzs	Elzonris^TM^	fusion protein	Blastic plasmacytoid dendritic cell neoplasm
Tildrakizumab	Ilumya^TM^	monolclonal antibody	Moderate-to-severe plaque psoriasis

^a^ by alphabetical order; ^b^ USA.

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
