# Peer review of "The Pharmaceutical Industry in 2018. An Analysis of FDA Drug Approvals from the Perspective of Molecules"

_molecules, 2019, doi:10.3390/molecules24040809_

Round 1
Reviewer 1 Report
This is a nice review as the ones the authors did for the previous two years. But I have a few suggestions for the authors to consider to improve its readability.
Divide the manuscript into Introduction, Analysis, Discussion, and Conclusion.
Add a pie chart with all categories for these approved drugs like the authors did in their previously published article titled 2017 FDA Peptide Harvest in MDPI Pharmaceuticals.
Add manufacturer info for each approved drug.
Add a column for the info in foot notes on symbols in Table 1.
Capitalize the first letters of drugs in Figures 17 and 19.
Delete 242. The structures in Figure 18 seem related.
Rephrase lines 185-186, 249-250, and 277-280.
Author Response
Referee 1
This is a nice review as the ones the authors did for the previous two years. But I have a few suggestions for the authors to consider to improve its readability.
Thank you very much for your kind words.
Divide the manuscript into Introduction, Analysis, Discussion, and Conclusions.
We have divided in Analysis, Discussion, and Conclusions. In our opinion, there is not Introduction.
Add a pie chart with all categories for these approved drugs like the authors did in their previously published article titled 2017 FDA Peptide Harvest in MDPI Pharmaceuticals.
We have already did it. Thanks for the suggestion.
Add manufacturer info for each approved drug.
This information is very difficult to find. In recent years, mostly of the new IND are developed by a company, many often a biotech, then it is sold to a big company, and finally the manufacturers (at least two) are CMO.
Add a column for the info in foot notes on symbols in Table 1.
We did it. Again, thanks.
Capitalize the first letters of drugs in Figures 17 and 19.
Done
Delete 242. The structures in Figure 18 seem related.
Done
Rephrase lines 185-186, 249-250, and 277-280.
Done
Reviewer 2 Report
This review article by de la Torre and Albericio provides an excellent account of FDA approved drugs during the year 2018. The authors, who also published a similar paper in 2017, do a good job at summarizing the main classes of molecules (with corresponding molecular structures) approved and for which specific indications. Overall, the paper is easy to read and follow and describes a topic that should be of interest to the readership of this journal.
Minor comments:
- increase resolution of Figure 1
- check manuscript throughout for typos/minor errors
Author Response
Referee 2
This review article by de la Torre and Albericio provides an excellent account of FDA approved drugs during the year 2018. The authors, who also published a similar paper in 2017, do a good job at summarizing the main classes of molecules (with corresponding molecular structures) approved and for which specific indications. Overall, the paper is easy to read and follow and describes a topic that should be of interest to the readership of this journal.
Thanks to the referee for nice comments.
Minor comments:
- increase resolution of Figure 1
Done
- check manuscript throughout for typos/minor errors
Done
Reviewer 3 Report
The manuscript provides interesting information on the chemical entities approved by FDA for diverse indications. A bar chart showing expansion in the molecular classes of approved drugs (TIDES, small molecules,biologics) over the previous years is recommended.
Author Response
Referee 3
The manuscript provides interesting information on the chemical entities approved by FDA for diverse indications. A bar chart showing expansion in the molecular classes of approved drugs (TIDES, small molecules,biologics) over the previous years is recommended.
Many thanks